# Primary Care Professionals’ Empathy and Its Relationship to Approaching Patients with Risky Alcohol Consumption

**DOI:** 10.3390/healthcare12020262

**Published:** 2024-01-19

**Authors:** Celia Pérula-Jiménez, Esperanza Romero-Rodríguez, Jessica Fernández-Solana, José Ángel Fernández-García, Juan Manuel Parras-Rejano, Luis Ángel Pérula-de Torres, Ana González-de la Rubia, Josefa González-Santos

**Affiliations:** 1Maimonides Biomedical Research Institute of Cordoba (IMIBIC), Reina Sofia University Hospital, University of Cordoba, 14011 Cordoba, Spain; celia.perula.sspa@juntadeandalucia.es (C.P.-J.); esperanza.romero.rodriguez.sspa@juntadeandalucia.es (E.R.-R.); jangel.fernadez.sspa@juntadeandalucia.es (J.Á.F.-G.); 2Montoro Health Center, Andalusian Health Service, 14600 Cordoba, Spain; 3Cordoba and Guadalquivir Health District, 14001 Cordoba, Spain; juanm.parras.sspa@juntadeandalucia.es; 4Department of Health Sciences, University of Burgos, 09001 Burgos, Spain; mjgonzalez@ubu.es; 5Villarrubia Center, 14005 Cordoba, Spain; 6Huerta de la Reina Health Center, Andalusian Health Service, 14600 Cordoba, Spain; 7Research Network on Chronicity, Primary Care and Prevention and Health Promotion (RICAPS-ISCIII), 08007 Zaragoza, Spain; luisangel.perula@gmail.com; 8Program of Preventive Activities and Health Promotion (PAPPS-semFYC), 08009 Barcelona, Spain; 9General Emergencies Unit, Regional University Hospital, 29010 Malaga, Spain; ana.gonzalez.rubia.sspa@juntadeanducia.es

**Keywords:** empathy, healthcare professionals, primary care, alcohol consumption, clinical performance, Jefferson Scale of Empathy

## Abstract

The aim of this study was to estimate the level of empathy among primary care (PC) health professionals and its relationship with their approach to patients at risk due to alcohol consumption. This is an observational, descriptive, and multicenter study that included 80 PHC professionals. The professionals completed a questionnaire comprising socio-occupational questions and inquiries regarding their actions when dealing with patients suspected of risky alcohol consumption. The Jefferson Scale of Empathy was used to measure their level of empathy and was completed by 80 professionals, of whom 57.5% were family physicians, 10% were nurses, and 32.5% were family- and community-medicine residents. The mean age was 39.5 ± 13.1 (SD) (range of 24–65 years) and 71.3% were females. The mean empathy level score was 112.9 ± 11.1 (95% CI: 110.4–115.4; range: 81–132 points). Actions that stood out for their frequency were providing health advice in the general population, offering advice to pregnant women, and recommending abstinence to users of hazardous machinery or motor vehicles. The level of empathy was associated with age (*p* = 0.029), the health center’s scope (*p* = 0.044), systematic alcohol exploration (*p* = 0.034), and follow-ups for patients diagnosed with risky consumption (*p* = 0.037). The mean score obtained indicated a high level of empathy among professionals. Professionals with greater empathy more frequently conducted systematic screening for risky alcohol consumption.

## 1. Introduction

The use of alcohol constitutes one of the main preventable reasons for illnesses and deaths, being a key risk factor for non-communicable diseases such as cardiovascular diseases, liver cirrhosis, and various types of cancer [1]. Currently, harmful alcohol consumption significantly contributes to the global burden of disease, accounting for approximately 5.1% of the global burden of morbidity and injuries, calculated in terms of disability-adjusted life years (DALYs) [2,3].

Alcohol, a psychoactive substance linked to hazardous consumption and dependency, is affected by both the frequency of use and the quantity consumed, which impact its effects [4]. Harmful alcohol consumption results in a significant increase in the utilization of health services, both at the hospital level and in primary care (PC), accounting for between 15% and 20% of the consultations attended by family doctors in this latter setting [5].

Currently, the criteria used to define risky consumption patterns are clearly outlined in the Program of Preventive Activities and Health Promotion (PAPPS) [6]. This program, supported by the Spanish Society of Family and Community Medicine (semFYC), stands as a reference for prevention using PC in Spain. In this context, a distinction is made between habitual consumers (risk drinkers) and intensive consumers (binge drinkers) [7,8].

To understand alcohol consumption patterns, it is essential to consider the standard drink unit (SDU), which is the standard unit for quantifying alcohol [9]. An SDU equals 8–10 g of pure alcohol, with 10 g being the most practical and commonly used measure in Europe. The formula used to calculate the SDU is as follows: SDU = volume in liters multiplied by the alcohol percentage in the drink, divided by 0.8 (since one milliliter of alcohol contains 0.785 g of alcohol).

Risky alcohol consumption is characterized by intake levels that can be harmful to the body, corresponding to an amount exceeding 17 standard drink units (SDUs) per week or 2–2.5 SDUs per day in women, and more than 28 SDUs per week or 4 SDUs per day in men. In contrast, low-risk consumption involves an alcohol intake of less than 17 SDUs per week and 2–2.5 SDUs per day in women, as well as fewer than 28 SDUs per week and 4 SDUs per day in men. Intensive alcohol consumption, known as binge drinking, is defined as the intake of more than 6 SDUs in a short period [10,11].

According to data from the National Statistics Institute (INE), the daily alcohol consumption in the autonomous community of Andalusia (Spain) is 11.32 g per drink unit. Similarly, based on social class as determined by the occupation of the reference person, healthcare professionals have an average weekly consumption of 10.31 g per drink unit, with a consumption of 5.41 g during the weekdays (Monday to Thursday) and 16.89 g on weekends (Friday to Sunday) [12,13].

PC health professionals play a crucial role in managing alcohol-related issues, as they constitute the first line of healthcare. Their comprehensive approach includes patient care, family involvement, and the implementation of health-promotion and disease-prevention activities [14].

The actions of PC health professionals regarding harmful alcohol consumption involve actively seeking cases and applying intervention techniques aimed at modifying patients’ consumption patterns. The early identification of risky consumption through systematic screening is fundamental for preventive interventions in the healthcare setting. PC professionals, particularly nurses, play a crucial role in identifying and providing brief interventions to consumers at risk of alcohol consumption [15,16,17,18].

In this case, it is crucial to consider empathy as a necessary quality to provide quality care in clinical practice, given its clear impact on health outcomes. The interaction between a professional and a patient is considered a significant clinical indicator for delivering high-quality care [15]. Empathy can be defined as a professional’s ability to understand the perspectives, feelings, and situations of their patients [19]. Professionals’ empathy generates patient satisfaction, facilitating clinical decision making in a more genuine and logical manner. This indirectly contributes to better treatment adherence and outcomes [19,20,21,22,23]. Consequently, some researchers consider empathy a central competence for therapists treating risky drinking [24].

Empathy and empathetic communication play fundamental roles in the medical field, particularly when dealing with patients who exhibit risky alcohol consumption. These skills not only enable a deeper connection between healthcare professionals and patients, but also facilitate a more effective and personalized medical approach [25]. By understanding patients’ experiences, concerns, and needs, professionals can establish stronger, trusting relationships, which are crucial for the success of interventions and treatments related to alcohol consumption. Empathy and compassionate communication can help overcome barriers, foster openness in the doctor–patient relationship, and promote positive changes in the health and well-being of those facing alcohol-related issues [26,27].

Over the years, various questionnaires have been developed to assess empathy across all levels of medical professionals. However, the Jefferson Scale of Empathy (JSE) stands out as an essential element in healthcare due to its significant relevance in the healthcare professional–patient relationship [28].

Furthermore, there is little knowledge about the empathy level of PC professionals and whether it influences their approach to patients with suspected risky alcohol consumption. Therefore, the objective of this study was to estimate the empathy level of PC health workers and analyze its relationship with their approach to patients with possible risky alcohol consumption.

## 2. Materials and Methods

### 2.1. Study Design and Population

This was a descriptive and multicenter observational study conducted in PC centers within the Andalusian Health Service located in the province of Córdoba, Spain. Professionals from the Córdoba and Guadalquivir Health District, part of the Andalusian Health Service, were invited to participate. The inclusion criteria for the study were: (1) being a PC health professional (family physician, nurse, or resident specialist in family and community medicine or nursing) and (2) providing informed consent to participate in the study. The exclusion criterion was refusal to participate in the study.

Also, for these professionals to be included in the study, the patients they attended to in their consultations also needed to meet inclusion criteria to assess the degree of empathy of the professionals and their actions towards these patients at risk of alcohol consumption. That is to say, there was a third inclusion criterion for healthcare professionals, which involved their patients meeting certain criteria as well. These criteria were: (1) having a risky consumption of alcohol: (a) consuming more than 17 standard drink units (SDUs) (170 g of alcohol per week) for women, (b) consuming more than 28 SDUs (280 g of alcohol per week) for men, or (c) patients engaging in “binge drinking” (excessive or intensive consumption, i.e., men consuming 6 SDUs or more or women consuming 4 SDUs or more in less than 2 h); (2) being at least 14 years old; and (3) providing informed consent to participate in a clinical trial [10].

The exclusion criteria for patients included: (1) a severe cognitive impairment (such as severe dementia or psychosis) and/or a terminal illness; (2) a lack of social support or unemployment; and (3) the coexistence of another substance dependence supervised by addiction specialists.

### 2.2. Procedure

For participant recruitment, the study was disseminated through the Multiprofessional Teaching Unit of Family and Community Care in the Córdoba and Guadalquivir Health District. All study-related information was sent via email to the participating professionals.

Once the study’s objectives were explained and professionals were encouraged to participate, they completed informed consent forms. Subsequently, they received training on managing patients with risky alcohol consumption and provided standard clinical care and advice to the patients they attended to.

Participant information was obtained through a self-administered questionnaire designed by the study’s researchers, who were experts in addressing alcohol consumption in PC.

The research project obtained authorization from the management of the Córdoba and Guadalquivir Health District and approval from the Clinical Research Ethics Committee of the Reina Sofía Hospital in Córdoba. Informed consent was obtained from the study participants, ensuring voluntariness and anonymity. Data handling was conducted in compliance with the European Data Protection Regulation and the Organic Law 3/2018 on personal data protection and the guarantee of digital rights.

An alpha risk of 0.95 for a precision of +/− 5 units in a two-sided test for a standard deviation of 15, estimated for the expected value of the Jefferson Scale of Empathy score, was accepted. A total of 31 subjects, randomly selected from the whole population, was required, assuming that such a population was equal to 240 subjects. A replacement rate of 0% was anticipated. The calculations were carried out using the GRANMO program (https://apisal.es/Investigacion/Recursos/granmo.html) (accessed on 19 January 2023).

### 2.3. Instruments

The questionnaire included several sections: one on socio-occupational data (age, gender, marital status, contractual relationship, profession, and years of work), another to measure the professional’s empathy level, and one regarding their approach to patients with potential risky alcohol consumption. The following is the link to the questionnaire: https://docs.google.com/forms/d/e/1FAIpQLSf9sq52HqTRieufmWS26IIlBmf8GzWQ5eJEHDvsFQqCZfaGww/viewform (accessed 2 January 2023).

The questionnaire was created ad hoc by experts in the design of health questionnaires, and it was subjected to a qualitative validation process (face, logic, or consensus validity). The questionnaire was related to the clinical practice of professionals and consisted of 10 single-answer questions focused on: the systematic exploration of alcohol consumption, health advice to reduce alcohol intake in the general population, pregnant women or drivers of dangerous machinery, and how health professionals approach patients with risky alcohol consumption in their usual practice.

To determine the empathy level, the Jefferson Scale of Empathy was used, validated in Spanish [23,29]. It consists of 20 Likert-type questions, where the professional responds on a seven-point scale ranging from “completely agree” (7 points) to “completely disagree” (1 point). There are 10 positively worded items and 10 negatively worded questions to prevent response automatism. In the negatively worded items, the scoring is inverted, so when the individual responds “completely disagree”, they receive 7 points. The scores can range from 20 to 140 points. The Spanish validation of the scale identified three dimensions: Questions 1 to 10 correspond with “perspective taking” (Dimension 1), Questions 11 to 17 correspond with “compassionate care” (Dimension 2), and Questions 18 to 20 correspond with “putting oneself in the patient’s shoes” (Dimension 3). The scores obtained on the Jefferson Scale allowed the professionals to be categorized into four empathy-level groups based on the estimated percentiles of the sample: very high (121 points or more), high (114–120), medium (113–108), and low (less than 108 points).

### 2.4. Statistical Analysis

Descriptive parameters were calculated (with confidence intervals estimated at 95% certainty—IC 95%), and the statistical relationship between the level of empathy and the professional’s actions towards patients with risky alcohol consumption was analyzed. The Mann–Whitney U test, the Kruskal–Wallis test, a chi-squared test, and Spearman’s correlation were used (the Shapiro–Wilk test was applied to the empathy level to verify its non-normal distribution), considering a significance level of *p* ≤ 0.05. The analysis was performed using the statistical software SPSS v.29.0 (IBM SPSS Statistics) and an online calculator for estimating the IC 95% of the qualitative or proportional data (https://statologos.com/intervalo-de-confianza-para-proporcion-poblacional/) (accessed on 19 January 2023).

## 3. Results

In Table 1, the socio-demographic characteristics of the studied professionals are presented. The mean age of the sample was 39.5 ± 13.1 (SD) (range: 24–65 years; 95% CI: 36.6–42.4) and 71.3% were females (95% CI: 61.3–81.2%). Among these 80 subjects, 57.5% were family physicians, 10.0% were nurses, and 32.5% were family- and community-medicine residents.

Table 2 displays the distribution of clinical actions carried out by healthcare professionals, categorized by the frequency of their execution (<35%, 35–64%, or >65%).

Among the clinical actions performed, those with a frequency of execution > 65% were “providing health advice in the general population”, “providing health advice to pregnant women”, and “advising abstinence to users of dangerous machinery or motor vehicles”, at 67.5%, 81.3%, and 57.5%, respectively. “Systematic alcohol exploration” and the “completion of any screening questionnaire” were the least-frequently performed actions, with a frequency of execution < 35% in 37.5% and 53.8% of cases, respectively.

As shown in Table 3, a significant association between age and the level of empathy was observed (*p* = 0.029). Professionals under 30 years old showed a prevalence of 38.7% for low empathy, while those over 45 years old exhibited a marked increase in high/very high empathy (62.1%).

The status of being a tutor of residents showed a significant trend (*p* = 0.052) toward higher empathy, with 70.6% exhibiting high/very high empathy among those who have been or are currently tutors.

The health center setting was significantly associated (*p* = 0.044) with the level of empathy. Professionals in urban centers showed a higher prevalence of high/very high empathy (61.4%) compared to those in rural areas, which had a prevalence of low empathy of 33.3%.

In Table 4, a significant relationship between the level of empathy and the systematic exploration of alcohol was observed (*p* = 0.034). Professionals with high/very high levels of empathy demonstrated a higher frequency of performing this exploration compared to those with low empathy.

Additionally, there was a significant relationship (*p* = 0.068) between a higher frequency of providing health advice to the general population and professionals with high/very high empathy compared to those with low empathy.

A significant relationship (*p* = 0.037) was also found between the level of empathy and the follow-up of patients diagnosed with risky alcohol consumption. Professionals with high/very high empathy showed a greater tendency to carry out this follow-up compared to those with low empathy.

Regarding the level of empathy demonstrated by the healthcare professionals in the sample, their distribution is depicted in Figure 1. A total of 20% of the sample exhibited a very high level of empathy, 28.75% showed a high level, 23.75% demonstrated a moderate level, and finally, 27.50% displayed a low level of empathy.

Significant differences were observed (*p* = 0.003) in the total scores for the action “systematic exploration of alcohol” and in Dimension 2 (compassionate care) of this action. Professionals who carried out systematic exploration of alcohol in more than 65% of cases obtained higher scores, 117.5 ± 7.11 (114.2–120.8), in these dimensions and in their overall empathy, followed by those who did so in 36–64% of cases (115.1 ± 11.4).

Significant associations were found for the total scores of “providing health advice in the general population” (*p* = 0.022) for those who provided advice in more than 65% of cases. These professionals demonstrated higher levels of empathy (114.9 ± 10.6), especially in Dimension 1 (perspective taking) (63.8 ± 7.0).

For the action “follow-up of patients diagnosed with risky alcohol consumption”, significant associations were observed in the total scores (*p* = 0.014) for those who conducted follow-ups in more than 65% of cases (119.2 ± 7.0). These professionals stood out in Dimension 1 (perspective taking) (12.8 ± 1.8), with a significant value of *p* = 0.013 (Table 5).

Figure 2 and Figure 3 display the average level of empathy among healthcare professionals, broken down by specific dimensions and the total score. Each dimension was evaluated using the established scale, where Dimension 1 is related to perspective taking, Dimension 2 is associated with compassionate care, and Dimension 3 is centered around the ability to put oneself in the patient’s shoes.

In Figure 2, according to the average score per question obtained by professionals in specific dimensions and the total score, it was noticeable that Dimension 1 received the highest rating, with an average of 6.28 points per question (range: 1–7), followed by Dimension 2, with an average of 5.50 per question, and finally, by Dimension 3, with an average of 3.89.

In Figure 3, the total average scores obtained in each dimension are presented, representing the sum of all the questions each dimension comprised. For Dimension 1, consisting of 10 questions, the average score obtained was 62.75 (range: 10–70). Dimension 2, composed of seven questions, had an average score of 38.48 (range: 7–47), and Dimension 3, constituted by three questions, yielded an average score of 11.68 (range: 3–21). These figures reflect that the overall mean score on the scale was 112.9 (range: 20–140), translating to a very high score according to this scale.

## 4. Discussion

The aim of this study was to assess the level of empathy among PC healthcare providers and analyze its relationship with their approach to patients with a potential risk of alcohol consumption.

Preventive interventions in alcohol consumption within the healthcare sphere play a crucial role in promoting health and preventing associated disorders. These strategies encompass the early detection of problematic consumption, personalized guidance, and counseling. They also involve educating people on the risks of excessive drinking, promoting healthy lifestyles, and advocating strategies to reduce harmful alcohol use. These interventions aim not only to identify and address risky consumption, but also to offer support and resources to those affected by its adverse effects, thus contributing to the health and well-being of the community [30,31].

Additionally, empathy plays a pivotal role in addressing patients with risky alcohol consumption by healthcare professionals. This quality enables them to understand and connect with patients’ experiences, concerns, and needs in a deeper and more meaningful way. By displaying empathy, professionals can establish a stronger and more trusting relationship with patients, facilitating open communication and a patient’s willingness to seek help and follow recommended treatments. Moreover, empathy contributes to creating a more comprehensive and collaborative care environment, which is crucial for the effectiveness of interventions and patient motivation towards positive changes in health and lifestyle [32,33,34].

Our results highlight that, among the clinical actions performed by professionals, advising the general population, pregnant women, and individuals operating dangerous machinery or motor vehicles were carried out in over 65% of cases. However, in our overall analysis of the clinical practice reported by healthcare providers, figures below 40% were observed regarding the systematic exploration of alcohol consumption in PC consultations, with a frequency of less than 35%, despite it being a key cornerstone of preventive alcohol interventions in healthcare settings [35,36]; this was also reflected in the completed screening questionnaires.

Our findings also revealed a significant association between healthcare professionals’ level of empathy and their age, with professionals under 30 years old exhibiting a higher prevalence of low empathy (38.7%), while those over 45 years old stood out for their high/very high levels of empathy (62.1%). Additionally, a significant association was established between the level of empathy and the healthcare center setting, with professionals in urban healthcare centers showing higher levels of empathy (61.4%).

Based on this study’s findings, it is evident that the actions taken in addressing patients at risk of alcohol consumption are directly linked to the healthcare professional’s level of empathy. Specifically, those displaying higher empathy tended to excel in providing healthcare advice to the general population, pregnant women, and individuals operating dangerous machinery or motor vehicles. Conversely, individuals with lower levels of empathy tended to lack in systematically exploring alcohol consumption, completing screening questionnaires, and following up with diagnosed patients at risk. This highlights the importance of tailoring communication strategies to the specific needs of each population group, meaning that they should be adjusted based on their culture or level of education. The aim is to achieve a higher level of empathy and employ appropriate communication strategies with the patient, ultimately fostering a more consistent professional–patient relationship and achieving greater success in treatments.

Additionally, this study examined the mean empathy scores of healthcare professionals, broken down by dimensions and overall, in relation to their actions in addressing patients with risky alcohol consumption. The results revealed statistically significant associations in several evaluated dimensions. Specifically, significant differences were observed in Dimension 2, regarding the systematic exploration of alcohol and its total, as well as in providing healthcare advice to the general population in Dimension 3 and the total for patient follow-up in those with a diagnosed consumption risk.

Collectively, this study provided a comprehensive understanding of healthcare professionals’ average level of empathy, offering valuable insights into how this trait manifests in different key dimensions. Dimension 1 scored higher compared to the other two dimensions, indicating that healthcare professionals have better perspective-taking abilities, but might struggle more with putting themselves in the patient’s shoes (Dimension 3). Overall, they demonstrated a high level of empathy concerning their approach to patients with risky consumption behavior.

According to the National Institute on Alcohol Abuse and Alcoholism in the United States, regular consultations with primary healthcare professionals that focus on addressing alcohol consumption can significantly improve the condition of patients dealing with harmful consumption [37]. However, a lack of understanding among professionals on how to handle alcohol consumption has been identified as one of the main barriers in their consultations. Previous research by Nilsen et al. [38] and Johnson et al. [39] highlighted a deficient level of knowledge among general physicians and nurses concerning alcohol. When assessing fundamental concepts in alcohol treatment such as UBE (use, brief intervention, and engagement), risky drinking, and occasional excessive drinking, a low percentage of correct responses was observed, ranging between 35% and 52%.

In Spain, various local studies have identified the level of knowledge among healthcare professionals regarding alcohol consumption. A study in Catalonia revealed a low level of knowledge in addressing alcohol consumption, primarily due to a lack of training in the prevention of alcohol consumption [40].

Currently, several studies have demonstrated that healthcare advice can achieve significant and sustained reductions in alcohol consumption. Ballesteros et al. [41], in their meta-analysis of Spanish studies in PC, highlighted the efficacy of medical advice in reducing alcohol consumption. Moreover, international studies such as Bertholet et al.’s trial [42] have concluded that brief interventions are effective in reducing alcohol consumption at 6 and 12 months of follow-up, even for more extended periods.

Brief interventions providing medical advice have proven to be highly effective in reducing the alcohol consumption among individuals with risky drinking patterns, while also mitigating the associated morbidity and mortality. Even for patients who refuse referrals, regular visits to primary healthcare professionals that are focused on addressing alcohol dependence can lead to notable improvements. According to the World Health Organization (WHO), a brief intervention stands as one of the most cost-effective strategies in PC concerning alcohol consumption, ranking second only to tobacco interventions [43,44,45,46,47].

This underscores the importance of following clinical practice guidelines, which are backed by robust scientific evidence supporting alcohol screening in pregnant women [48] and individuals operating dangerous machinery or driving vehicles, given the severe risks involved [49].

However, several obstacles hinder healthcare professionals’ interventions in alcohol-related issues [50]. These include time constraints in addressing this problem, inadequate training, negative attitudes of PC staff towards patients with unfavorable prognoses, a limited knowledge of treatment effectiveness, and rejection associated with prejudices [39,51].

Nevertheless, various studies have revealed that empathy can vary among professionals and be influenced by factors such as personal stability, their prioritization of others’ well-being, the relationship between religion and empathy, cultural influences, and childhood education. These elements explain variations, both among individuals and within the same individual, in terms of their experiences and expressions of empathy [52].

This underscores the need to implement specific programs aimed at enhancing empathy in clinical settings. Specialists in alcohol treatment highlight that more positive attitudes among clinical professionals are linked to increased intervention activity. Moreover, training and support targeted at these professionals are associated with positive changes in their attitudes and a greater willingness to intervene [53].

Different educational initiatives conducted in Spain, such as the one led by the team of Ruiz Moral et al. [54], have demonstrated how training in this field generates positive perceptions and attitudes in crucial aspects of clinical interaction, influencing behavioral changes in professionals [55].

Currently, training for healthcare professionals on managing alcohol consumption is considered a fundamental tool in caring for patients with harmful drinking habits [56]. It is crucial to direct this training to all professionals involved in PC: family physicians, nursing staff, and trainee students.

Additionally, there is mounting evidence regarding the crucial role of nursing in detecting alcohol consumers with risky patterns and implementing brief interventions. It is recommended that the entire PC team participates in brief intervention programs, clearly defining the responsibilities of each professional. This would allow for a comprehensive assessment of patients and an individualized approach based on the particularities of their alcohol consumption and associated factors [16].

In one study, it was observed that a brief educational intervention for medical students using video modeling on screening, brief interventions, and referral for treatment (SBIRT) had an unexpected and lasting impact on their empathic communication skills. Apart from displaying greater empathy, the students were more inclined to suggest reducing alcohol consumption [57].

Through previous studies involving professionals, it was noted that the variation in empathy among therapists was related to the outcomes, as small increases in a therapist’s baseline level of empathy were associated with more significant reductions in alcohol consumption at the end of treatment [58].

Similarly, research highlights that empathy develops over time through experience, suggesting that older professionals might have higher levels of empathy. It underscores the need for experiential-based teaching approaches for students to nurture their empathy. In an increasingly digital world, a simulated environment can be provided where students can practice empathy for their future roles [59].

Hence, both the level of education and the attitudes and communication skills present in professionals are fundamental elements in managing alcohol consumption. This was confirmed in a trial by Anderson, conducted in five European countries, which emphasized that training is essential for acquiring an appropriate attitude in handling alcohol [60]. Furthermore, Rosário stated that professionals with a more positive attitude have a better approach to patients with alcohol consumption issues [61].

These findings underscore the importance of considering and fostering empathy in training and clinical practice to enhance the quality of care for patients with alcohol-related risky consumption. In other words, fostering empathy among healthcare professionals treating patients with risky alcohol consumption is crucial to ensure effective and compassionate care. Empathy enables healthcare professionals to connect more closely with their patients’ experiences and needs, fostering a more human and understanding care environment. To strengthen this aspect, specific interventions could be implemented, such as training programs in empathic communication, empathic care environments, mentorship and supervision, regular empathy assessments with feedback, support and reflection groups, and best practices included in practical training.

However, this study has several limitations. The sample selection was limited to a specific group of healthcare professionals in a particular geographic location. Being a pilot study, the results may not be generalizable to the wider population. This may affect the generalizability of the results to other populations and contexts, suggesting the need for studies with more diverse and representative samples. Additionally, the measurement of empathy relied on the perception and self-reporting of healthcare professionals. This may have introduced biases due to the subjectivity of the responses and the possibility of socially desirable responses.

As for future research directions, we are continuing this line of investigation by collecting nationwide data from healthcare professionals working with patients engaging in risky alcohol consumption. Furthermore, there is a plan to investigate the effectiveness of specific training programs designed to enhance the empathy and communication skills of healthcare professionals in alcohol-related situations. Exploring the underlying causes that may influence the levels of empathy could provide valuable insights for designing targeted interventions.

## 5. Conclusions

In summary, the results indicate that healthcare professionals commonly offer health guidance to the general public and pregnant women, and advise caution to individuals operating hazardous machinery or motor vehicles. Yet, less-commonly executed actions involve systematically probing alcohol consumption and filling out detection questionnaires.

Additionally, a significant association was observed between the age of the healthcare professional or the healthcare center’s environment and the levels of empathy among professionals. Moreover, a significant relationship was noted between the demonstrated level of empathy among professionals and the performance of systematic alcohol exploration, as well as the follow-up of patients diagnosed with alcohol consumption.

These results underscore the importance of ongoing training in communication skills and empathy for healthcare professionals. This continuous education can significantly contribute to more effective and patient-centered care, particularly in situations associated with alcohol consumption. By enhancing their communication and empathetic abilities, healthcare providers can establish a stronger rapport, foster trust, and better understand the needs and concerns of patients dealing with alcohol-related issues. This, in turn, can lead to more tailored and supportive interventions, ultimately improving patient outcomes and the overall quality of care provided.

## Figures and Tables

**Figure 1 healthcare-12-00262-f001:**
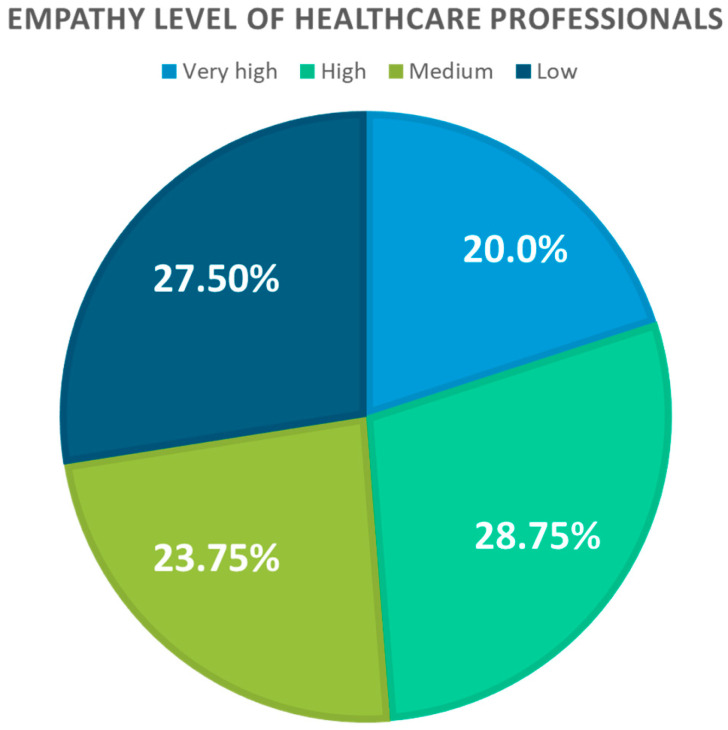
Level of empathy among healthcare professionals.

**Figure 2 healthcare-12-00262-f002:**
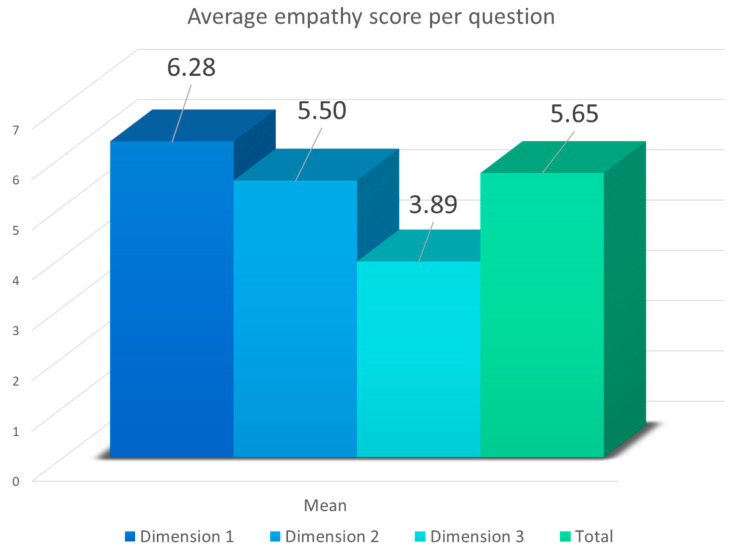
Average score per question (range 1–7) of empathy among healthcare professionals based on dimensions and total score (Dimension 1: perspective taking; Dimension 2: compassionate care; Dimension 3: putting oneself in the patient’s shoes).

**Figure 3 healthcare-12-00262-f003:**
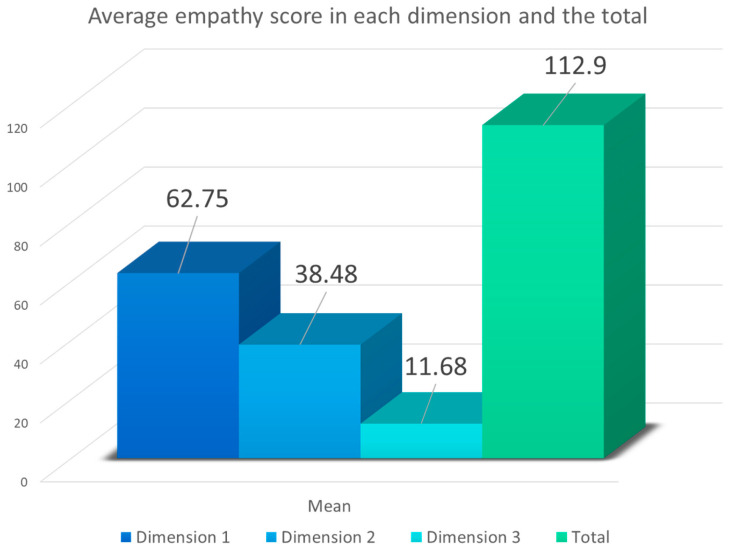
Average empathy scores of healthcare professionals according to dimensions (Dimension 1: perspective taking (range: 10–70); Dimension 2: compassionate care (range: 7–49); Dimension 3: putting oneself in the patient’s shoes (range: 3–21)) and the total score (range: 20–140).

**Table 1 healthcare-12-00262-t001:** Socio-demographic characteristics of the studied professionals.

Variables		Total N = 80		
		*n*	%	IC 95%
Sex	Male	23	28.8%	18.8–38.7
	Female	57	71.3%	61.3–81.2
Age	<30 years	31	38.8%	28.1–49.4
	30–45 years	20	25%	15.5–34.5
	>46 years	29	36.3%	25.8–46.8
Marital status	Married	37	46.3%	35.4–57.2
	Divorced	1	1.3%	0.6–2.0
	Separated	1	1.3%	0.6–2.0
	Single	39	48.8%	37.8–59.7
	Widowed	2	2.5%	1.6–3.4
Profession	Nurse	8	10.0%	3.4–16.5
	PC physician	46	57.7%	46.9–68.5
	Resident physician	26	32.5%	22.2–42.8
Contract type	In training	34	42.5%	31.7–53.3
	Temporary	13	16.3%	8.2–24.4
	Interim	5	6.3%	0.1–11.6
	Owner	28	35.0%	24.5–45.4
Work environment	Rural	5	6.3%	0.1–11.6
	Semi-urban	31	38.8%	28.1–49.5
	Urban	44	55.0%	44.1–65.9
Residency tutor	Yes	17	21.3%	12.3–30.3
	No	63	78.8%	69.8–87.8
Years of work experience	<6 years	39	48.8%	38.8–59.7
	6–15 years	19	23.8%	14.5–33.1
	16–25 years	11	13.8%	6.2–21.6
	>25 years	11	13.8%	6.2–21.6

**Table 2 healthcare-12-00262-t002:** Clinical performance of healthcare professionals in addressing patients with risky alcohol consumption.

Clinical Performance	Execution Frequency	Total N = 80		
		*n*	%	95% CI
Systematic alcohol exploration	<35%	30	37.5	26.9–48.1
	35–64%	30	37.5	26.9–48.1
	>64%	20	25.0	15.5–34.5
Completion of any screening questionnaires	< 35%	43	53.8	42.9–64.7
	35–64%	17	21.3	12.3–30.2
	>64%	20	25.0	15.5–34.5
Providing health advice in the general population	<35%	12	15.0	7.2–22.8
	35–64%	14	17.5	9.2–25.8
	>64%	54	67.5	57.2–77.8
Providing health advice to pregnant women	<35%	13	16.3	8.2–24.4
	35–64%	2	2.5	0.0–5.9
	>64%	65	81.3	72.7–89.8
Advising abstinence to users of dangerous machinery or motor vehicles	<35%	21	26.3	16.7–36.9
	35–64%	13	16.3	8.2–24.4
	>64%	45	57.5	46.6–68.3

**Table 3 healthcare-12-00262-t003:** Level of empathy based on socio-demographic and occupational characteristics of healthcare professionals.

Variables		Level of Empathy Total (*n* = 80)						*p*-Value
		Low		Medium		High/Very High		
		*n*	%	*n*	%	*n*	%	
Age	<30 years	12	38.7%	8	25.8%	11	35.5%	0.029 *
	30–45 years	5	25.0%	5	25.0%	10	50.0%	
	>45 years	5	17.25	6	20.7%	18	62.1%	
Sex	Male	10	43.5%	3	13.0%	10	43.5%	0.094
	Female	12	21.1%	16	28.1%	29	50.9%	
Current personal relationship	With partner	13	33.3%	10	25.6%	16	41.0%	0.165
	Without partner	9	22.0%	9	22.0%	23	56.1%	
Profession	Nurse	2	25.0%	3	37.55%	3	37.5%	0.287
	PC physician	10	21.7%	19	23.85%	27	58.7%	
	Resident physician	10	38.5%	7	26.9%	9	34.6%	
Contract type	In training	13	38.2%	8	23.55%	1	20.0%	0.208
	Temporary	10	32.3%	6	33.3%	8	44.4%	
	Owner	5	17.95%	5	17.9%	18	64.3%	
Residency tutor	Yes (is or has been)	4	23.5%	1	5.95%	12	70.6%	0.052
	No	18	28.6%	18	28.6%	27	42.9%	
Work environment	Urban ^a^	10	22.7%	7	15.9%	27	61.4%	0.044 *
	Semi-rural/rural ^b^	12	33.3%	12	33.3%	12	33.3%	

^a^: >10,000 inhabitants; ^b^: 10,000 inhabitants or fewer; * *p* < 0.05.

**Table 4 healthcare-12-00262-t004:** The relationship between the level of empathy and the healthcare professional’s performance in addressing risky alcohol consumption.

Clinical Performance	Execution Frequency	Level of Empathy Total (*n* = 80)						*p*-Value
		Low		Medium		High–Very High		
		*n*	%	*n*	%	*n*	%	
Systematic alcohol exploration	<35%	14	63.6%	8	42.1%	8	20.5%	0.034 *
	36–64%	6	27.3%	5	26.3%	19	48.7%	
	>64%	2	9.1%	6	31.6%	12	30.8%	
Completion of any screening questionnaires	<35%	13	59.1%	12	63.2%	18	46.2%	0.691
	36–64%	5	22.7%	3	15.8%	9	23.1%	
	>64%	4	18.2%	4	21.1%	12	30.8%	
Providing health advice in the general population	<35%	2	9.1%	4	21.1%	6	15.4%	0.068
	36–64%	8	36.4%	3	15.8%	4	7.7%	
	>64%	12	54.5%	12	63.2%	30	76.9%	
Providing health advice to pregnant women	<35%	4	18.2%	1	5.3%	8	20.5%	0.569
	36–64%	1	4.5%	0	0.0%	1	2.6%	
	>64%	17	77.3%	18	94.7%	30	76.9%	
Advising abstinence to users of dangerous machinery or motor vehicles	<35%	5	22.7%	3	15.8%	13	33.3%	0.483
	36–64%	5	22.7%	2	10.5%	6	15.4%	
	>64%	12	54.5%	14	73.7%	20	51.3%	
Conducting follow-ups with patients diagnosed with risky alcohol consumption	<35%	16	72.7%	12	63.2%	15	38.5%	0.037 *
	36–64%	5	22.7%	3	15.8%	13	33.3%	
	>64%	1	4.5%	4	21.1%	11	28.2%	

* *p* < 0.05.

**Table 5 healthcare-12-00262-t005:** Average scores obtained for the level of empathy, by dimensions and overall, according to the actions of professionals in addressing patients with risky alcohol consumption.

Clinical Performance	Execution Frequency	Dimension 1 Mean ± SD (95% CI)	*p*	Dimension 2 Mean ± SD (95% CI)	*p*	Dimension 3 Mean ± SD (95% CI)	*p*	Total Mean ± SD (95% CI)	*p*
Systematic alcohol exploration	<35%	60.5 ± 7.5 (57.6–63.3)	0.070	36.4 ± 5.4 (34.4–38.4)	0.007 *	10.8 ± 3.1 (9.6–11.9)	0.055	107.6 ± 11.3 (103.4–111.8)	0.003 *
	36–64%	63.5 ± 7.8 (60.6–66.5)		39.4 ± 4.0 (37.9–40.9)		12.1 ± 2.4 (11.2–13.0)		115.1 ± 11.4 (110.8–119.3)	
	>64%	65.0 ± 4.7 (62.8–67.2)		40,1 ± 3.7 (38.4–41.9)		12.4 ± 1.8 (11.5–12.3)		117.5 ± 7.11 (114.2–120.8)	
Providing health advice in the general population	<35%	63.0 ± 7.9 (58.1–68.1)	0.081	37.0 ± 4.1 (34.4–39.6)	0.181	11.7 ± 2.6 (10.1–13.4)	0.054	111.8 ± 10.8 (105.0–118.6)	0.022 *
	36–64%	58.9 ± 6.9 (54.8–62.9)		36.9 ± 4.7 (34.2–39.7)		10.1 ± 2.7 (8.6–11.7)		105.9 ± 11.1 (99.5–112.3)	
	>64%	63.8 ± 7.0 (61.8–65.6)		39.2 ± 4.8 (37.9–39.5)		12.1 ± 2.6 (11.3–12.7)		114.9 ± 10.6 (112.0–117.8)	
Monitoring patients diagnosed with risky alcohol consumption	<35%	61.6 ± 7.1 (59.4–63.8)	0.053	37.5 ± 5.0 (36.0–39.0)	0.143	10.9 ± 3.0 (10.0–11.8)	0.013 *	110.0 ± 11.4 (106.5–113.5)	0.014 *
	36–64%	62.1 ± 8.2 (58.4–65.9)		39.4 ± 3.9 (37.7–41.2)		12.4 ± 2.0 (11.5–13.3)		114.0 ± 11.3 (108.8–119.1)	
	>64%	66.6 ± 5.0 (63.9–69.3)		39.8 ± 4.8 (37.4–39.5)		12.8 ± 1.8 (11.9–12.3)		119.2 ± 7.0 (115.2–122.9)	

Dimension 1: perspective taking; Dimension 2: compassionate care; Dimension 3: putting oneself in the patient’s shoes. Note: Only actions with statistically significant associations are included. * *p* < 0.05.

## Data Availability

Data are contained within the article.

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
