# Peer review of "Primary Care Professionals’ Empathy and Its Relationship to Approaching Patients with Risky Alcohol Consumption"

_healthcare, 2024, doi:10.3390/healthcare12020262_

Round 1
Reviewer 1 Report
Comments and Suggestions for Authors
First of all, I would like to thank the editor for trusting my criteria for the evaluation of this article.
However, the theoretical framework presented is scarce and does not touch on the fundamental points of the research to be carried out. I believe that it is necessary for researchers to solidly substantiate the state of the art with other studies and significantly improve this section.
Methodologically we lack data, for example we cannot determine if the sample is significant because we do not have the total N to address this issue in the study if we have n=80 and 98 centers we have a representativeness of 1 per center and it is scarce to extrapolate the data and draw real conclusions. We also do not have any data on the patient approach questionnaire, it is said that it is done by experts but there is no description or discussion of its reliability and validity.
Although it is true that the results are to be expected in relation to similar studies, I still consider that this study does not have a representative sample that would allow the scientific community to draw conclusions that can be extrapolated to other contexts.
Author Response
Mrs. Jessica Fernández Solana
Department of health sciences
University of Burgos, Paseo Comendadores s/n.
Burgos, 09001, Spain
Tel. (+34) 947499108
Email: [email protected]
05-01-2024
Healthcare. Subject: Submissions Needing Revision
Dear editor.
Thank you very much for inviting us to submit our response to reviewers for our manuscript (healthcare-2782437) entitled: “Primary Care Professionals' Empathy and Its Relationship with Approaching Patients with Risky Alcohol Consumption.”
We have checked our manuscript according to the Academic Editor, the reviewers’ comments and the Journal requirements. We have also responded to some comments from reviewers point by point).
We would be very grateful if you could consider our manuscript to be published in your journal.
Yours sincerely,
Jessica Fernández Solana, OT, PT
- Response to Reviewer 1:
First of all, we would like to express our sincere gratitude for all comments and suggestions received from the Reviewer 1. This information has certainly enriched the text for its best understanding, thank you very much indeed. We have clarified the reviewer1’s questions. We have introduced the required changes both in our answers to the specific comments and in the final manuscript V2.
First of all, I would like to thank the editor for trusting my criteria for the evaluation of this article.
However, the theoretical framework presented is scarce and does not touch on the fundamental points of the research to be carried out. I believe that it is necessary for researchers to solidly substantiate the state of the art with other studies and significantly improve this section.
Response: Thank you for your comment, we have tried to improve this section by introducing some additional data and information.
Methodologically we lack data, for example we cannot determine if the sample is significant because we do not have the total N to address this issue in the study if we have n=80 and 98 centers we have a representativeness of 1 per center and it is scarce to extrapolate the data and draw real conclusions. We also do not have any data on the patient approach questionnaire, it is said that it is done by experts but there is no description or discussion of its reliability and validity.
Response: Thank you very much for your comment. However, what you say about the 98 centres is not correct, this figure has not been detailed in our manuscript, it is not correct. In this study, data was collected from 80 health professionals in various centres located in the Cordoba and Guadalquivir District. Furthermore, this study was considered a pilot study carried out in a very specific area of Spain, and as a future line of research, it will continue to be investigated with a larger sample in different parts of Spain. The latter was not detailed in the correct way, so this information has been added to the manuscript. The population scope was the health professionals assigned to the Multiprofessional Teaching Unit for Family and Community Care of the Córdoba-Guadalquivir Health District. At the time of recruitment, the unit was made up of 94 resident tutors, 32 teaching collaborators and 104 family and community medicine or nursing residents (N=240), located in 29 health centers (and not 98, as the reviewer notes). Taking this into account, the response rate was 33.3%.
This rate can be considered acceptable for this type of online studies by self-administered survey.
Information on the questionnaire used has also been added. (lines 167-172). Attached is the link to the questionnaire: https://docs.google.com/forms/d/e/1FAIpQLSf9sq52HqTRieufmWS26IIlBmf8GzWQ5eJEHDvsFQqCZfaGww/viewform.
The questionnaire was created ad hoc by experts in the design of health questionnaires, and it was subjected to a qualitative validation process (face, logic or consensus validity).
Although it is true that the results are to be expected in relation to similar studies, I still consider that this study does not have a representative sample that would allow the scientific community to draw conclusions that can be extrapolated to other contexts.
Response: Thank you for your comment. As we have commented previously, we have obtained a sample that may be representative of a small area such as the Cordoba and Guadalquivir Health District, however, extrapolating these results to the rest of the population is a limitation. This study was part of a pilot study which will be continued at a national level, this has been detailed more specifically in the manuscript for clarification and specification.
It is complex to determine the point from which we can affirm that the sample is representative or not; there are no agreed upon epidemiological criteria in relation to this issue. We estimate that a response rate of 33.3% is above what is usually common in this type of studies.
On the other hand, it is not easy to discern when a sample can be considered significant or not. We have made an estimate, and these are the results: “Accepting an alpha risk of 0.95 for a precision of +/- 5 units in a two-sided test for a standard deviation of 15 estimated in the expected value for the score with the test of Jefferson, 31 subjects randomly selected from the whole population are required assuming that such population is equal to 240 subjects. It has been anticipated a replacement rate of 0%”. (calculations carried out with the GRANMO program (https://apisal.es/Investigacion/Recursos/granmo.html) (see lines 156-160).
Thank you very much,
Jessica Fernández Solana, OT, PT

Reviewer 2 Report
Comments and Suggestions for Authors
In the conclusions, due to the small number of respondents, it is worth emphasizing that the study is of a pilot/preliminary nature and I inform about possible continuation. In my opinion, postulating conclusions are also missing, these studies show that more than half of the respondents have medium or low empathy. It is worth proposing actions to improve this situation.
Author Response
Mrs. Jessica Fernández Solana
Department of health sciences
University of Burgos, Paseo Comendadores s/n.
Burgos, 09001, Spain
Tel. (+34) 947499108
Email: [email protected]
05-01-2024
Healthcare. Subject: Submissions Needing Revision
Dear editor.
Thank you very much for inviting us to submit our response to reviewers for our manuscript (healthcare-2782437) entitled: “Primary Care Professionals' Empathy and Its Relationship with Approaching Patients with Risky Alcohol Consumption.”
We have checked our manuscript according to the Academic Editor, the reviewers’ comments and the Journal requirements. We have also responded to some comments from reviewers point by point).
We would be very grateful if you could consider our manuscript to be published in your journal.
Yours sincerely,
Jessica Fernández Solana, OT, PT
- Response to Reviewer 2:
First of all, we would like to express our sincere gratitude for all comments and suggestions received from the Reviewer 2. This information has certainly enriched the text for its best understanding, thank you very much indeed. We have clarified the reviewer2’s questions. We have introduced the required changes both in our answers to the specific comments and in the final manuscript V2.
In the conclusions, due to the small number of respondents, it is worth emphasizing that the study is of a pilot/preliminary nature and I inform about possible continuation. In my opinion, postulating conclusions are also missing, these studies show that more than half of the respondents have medium or low empathy. It is worth proposing actions to improve this situation.
Response: Thank you very much for your comment. We are sorry for any inconvenience caused by the reading. This study refers to a pilot study carried out in a small health district in the area of Cordoba and Guadalquivir, however it has a limitation that can be extrapolated to the rest of the population. We are counting on a future line of research that has already been launched to collect data at a national level. The pilot study nature of the study has been specified in the manuscript in order to clarify errors.
Also, the conclusions have been modified to try to improve the quality of this article and some actions have been proposed to improve this situation.
We hope we have now answered all your comments and we are looking forward to hearing from you again.
Thank you very much,
Jessica Fernández Solana, OT, PT

Reviewer 3 Report
Comments and Suggestions for Authors
This is an interesting study in which the authors explore the relationship between primary care professionals' empathy and the clinical approach to patients with risky alcohol consumption. Based on the results of the study, the authors support the importance of including empathy in the clinical training of healthcare professionals to ensure better clinical assistance to patients with risky alcohol consumption.
Major comments
Introduction
- The authors should provide data on the magnitude of alcohol consumption in the region in which the study was conducted. They should also show the relevance of the study for the healthcare system and healthcare provision at least in the region included in the study.
- The authors should provide information on the role of primary care professionals' empathy in approaching patients with risky alcohol consumption, as showed by literature data.
Material and methods
- In section 2.1, the authors mention inclusion criteria for patients who participated in the study. However, in the following sections, the authors refer only to the primary care professionals who participated in the study. Therefore, it must be clarified whether both patients and primary care professionals participated in this study or only primary care professionals.
- Section 2.3- The authors should describe in more detail the section of the questionnaire regarding the approach of the primary care professionals to patients with potential risky alcohol consumption.
Results
- The small number of participants may represent a significant limitation of this study. The authors should provide figures on the number of primary care professionals working in the region included in the study. They should also explain the small number of participants included in this study.
- Authors must provide information on how the clinical actions explored in this study were identified.
Discussion
- Discussions should focus more on the results of the study and relate them to the literature data. Also, more emphasis should be placed on the importance of the results obtained for specific interventions intended for patients with risky alcohol consumption.
Minor comments
- Lines 44- 45- The last part of the sentence is not clear. It seems that the word ”habit” is not properly used in that context. Please revise!
- Lines 85-86- not clear what the authors mean by ”therapist treating SUD” as SUD stands for Standard Drink Unit as previously mentioned. Please revise!
- Line 176- the word ”This” should be deleted.
- Line 206- the word ”trend” should be replaced with ”relationship”
- Line 311- It is not clear to which "each group" the authors refer.
Author Response
Mrs. Jessica Fernández Solana
Department of health sciences
University of Burgos, Paseo Comendadores s/n.
Burgos, 09001, Spain
Tel. (+34) 947499108
Email: [email protected]
05-01-2024
Healthcare. Subject: Submissions Needing Revision
Dear editor.
Thank you very much for inviting us to submit our response to reviewers for our manuscript (healthcare-2782437) entitled: “Primary Care Professionals' Empathy and Its Relationship with Approaching Patients with Risky Alcohol Consumption.”
We have checked our manuscript according to the Academic Editor, the reviewers’ comments and the Journal requirements. We have also responded to some comments from reviewers point by point).
We would be very grateful if you could consider our manuscript to be published in your journal.
Yours sincerely,
Jessica Fernández Solana, OT, PT
- Response to Reviewer 3:
First of all, we would like to express our sincere gratitude for all comments and suggestions received from the Reviewer 1. This information has certainly enriched the text for its best understanding, thank you very much indeed. We have clarified the reviewer1’s questions. We have introduced the required changes both in our answers to the specific comments and in the final manuscript V2.
Dear Authors,
Thank you for the opportunity to revise your manuscript. While the topic is relevant, there are serious concerns about the rational, study design, statistics and conclusions.
Please find below major issues:
This is an interesting study in which the authors explore the relationship between primary care professionals' empathy and the clinical approach to patients with risky alcohol consumption. Based on the results of the study, the authors support the importance of including empathy in the clinical training of healthcare professionals to ensure better clinical assistance to patients with risky alcohol consumption.
Major comments
Introduction
- The authors should provide data on the magnitude of alcohol consumption in the region in which the study was conducted. They should also show the relevance of the study for the healthcare system and healthcare provision at least in the region included in the study.
Response: Thank you very much for your comment. This information has been added in the introduction section.
- The authors should provide information on the role of primary care professionals' empathy in approaching patients with risky alcohol consumption, as showed by literature data.
Response: Thank you very much for providing this comment, I think it can add a quality point to the manuscript. We have tried to add information based on the literature about the role of empathy in health professionals in addressing risky consumption.
Material and methods
- In section 2.1, the authors mention inclusion criteria for patients who participated in the study. However, in the following sections, the authors refer only to the primary care professionals who participated in the study. Therefore, it must be clarified whether both patients and primary care professionals participated in this study or only primary care professionals.
Response: Thank you very much for your comment. We have tried to modify the wording to clarify the doubts about what was mentioned.
The inclusion criteria are twofold: on the one hand, the professionals had to meet specific inclusion criteria in order to be able to collect the data; and at the same time, in order for us to be able to collect data from these professionals who met the criteria, their patients also had to meet certain criteria. When both the professional and the patients met these criteria, the data could be collected for the study.
- Section 2.3- The authors should describe in more detail the section of the questionnaire regarding the approach of the primary care professionals to patients with potential risky alcohol consumption.
Response: Attached is the link to the questionnaire: https://docs.google.com/forms/d/e/1FAIpQLSf9sq52HqTRieufmWS26IIlBmf8GzWQ5eJEHDvsFQqCZfaGww/viewform.
The questionnaire related to the clinical practice of professionals consisted of 10 single-answer questions focused on: systematic exploration of alcohol consumption, health advice to reduce alcohol intake in the general population, pregnant women or drivers of dangerous machinery, and how health professionals approach patients with risky alcohol consumption in their usual practice.
Results
- The small number of participants may represent a significant limitation of this study. The authors should provide figures on the number of primary care professionals working in the region included in the study. They should also explain the small number of participants included in this study.
Response: It is complex to determine the point from which we can affirm that the sample is representative or not; there are no agreed upon epidemiological criteria in relation to this issue. We estimate that a response rate of 33.3% is above what is usually common in this type of studies.
On the other hand, it is not easy to discern when a sample can be considered significant or not. We have made an estimate, and these are the results: “Accepting an alpha risk of 0.95 for a precision of +/- 5 units in a two-sided test for a standard deviation of 15 estimated in the expected value for the score with the test of Jefferson, 31 subjects randomly selected from the whole population are required assuming that such population is equal to 240 subjects. It has been anticipated a replacement rate of 0%”. (calculations carried out with the GRANMO program (https://apisal.es/Investigacion/Recursos/granmo.html).
- Authors must provide information on how the clinical actions explored in this study were identified.
Response: This information was obtained from the socio-occupational and clinical practice questionnaire in which participants had to answer questions about the systematic exploration of alcohol consumption, and their empathy on this topic with the Jefferson Empathy Scale.
Discussion
- Discussions should focus more on the results of the study and relate them to the literature data. Also, more emphasis should be placed on the importance of the results obtained for specific interventions intended for patients with risky alcohol consumption.
Response: Thank you very much for your comment, changes have been made.
Minor comments
- Lines 44- 45- The last part of the sentence is not clear. It seems that the word ”habit” is not properly used in that context. Please revise!
- Lines 85-86- not clear what the authors mean by ”therapist treating SUD” as SUD stands for Standard Drink Unit as previously mentioned. Please revise!
- Line 176- the word ”This” should be deleted.
- Line 206- the word ”trend” should be replaced with ”relationship”
- Line 311- It is not clear to which "each group" the authors refer.
Response: Thank you very much, it has been modified.
We hope we have now answered all your comments and we are looking forward to hearing from you again.
Thank you very much,
Jessica Fernández Solana, OT, PT

Round 2
Reviewer 3 Report
Comments and Suggestions for Authors
I congratulate the authors and thank them for the effort to respond to my comments.
The authors have adequately addressed my comments and the manuscript can be considered for publication.